# Lifestyle and Progression to Type 2 Diabetes in a Cohort of Workers with Prediabetes

**DOI:** 10.3390/nu12051538

**Published:** 2020-05-25

**Authors:** Miquel Bennasar-Veny, Sergio Fresneda, Arturo López-González, Carla Busquets-Cortés, Antoni Aguiló, Aina M. Yañez

**Affiliations:** 1Department of Nursing and Physiotherapy, Balearic Islands University, Cra. de Valldemossa, Km 7,5, 07122 Palma, Illes Balears, Spain; miquel.bennasar@uib.es; 2Prevention of Occupational Risks in Health Services, Balearic Islands Health Service, C/Reina Esclaramunda, 9, 07003 Palma, Illes Balears, Spain; angarturo@gmail.com; 3Escuela Universitaria ADEMA, C/ Gremi de Passamaners, 11, 07009 Palma, Illes Balears, Spain; carla.busquests@uib.es; 4Research Group on Evidence, lifestyles and Health Research, Instituto de Investigación Sanitaria Illes Balears (IdISBa), Cra. de Valldemossa, Km 7,5, 07122 Palma, Illes Balears, Spain; aaguilo@uib.es (A.A.); aina.yanez@uib.es (A.M.Y.); 5Research Group on Global Health & Human Development, Balearic Islands University, Instituto de Investigación Sanitaria Illes Balears, Cra. de Valldemossa, Km 7,5, 07122 Palma, Illes Balears, Spain

**Keywords:** prediabetes, occupational health, lifestyles, risk factor, diabetes risk, fasting plasma glucose, HbA1c, normoglycemia, reversion

## Abstract

Background: People with prediabetes have an increased risk of developing type 2 diabetes (T2D). Few studies have evaluated the influence of lifestyle factors on the risk of progression to diabetes and reversion to normoglycemia. The aim of this study was to determine the incidence of T2D in a large cohort of workers with prediabetes, and to evaluate the influence of sociodemographic, clinical, metabolic, and lifestyle factors that affect the persistence of prediabetes and the progression to T2D. Methods: A cohort study of 27,844 adult workers (aged 20 to 65 years) from Spain who had prediabetes based on an occupational medical examination from 2012 to 2013. Prediabetes was defined as fasting plasma glucose (FPG) between 100 and 125 mg/dL. At the baseline evaluation, sociodemographic, anthropometric, metabolic, and lifestyle data were collected. At the 5-year follow-up, incident T2D was defined as an FPG of at least 126 mg/dL or initiation of an antidiabetic medication. Results: Among 235,995 initially screened workers, the prevalence of T2D was 14.19% (95% confidence interval (CI) 14.05 to 14.33) and the prevalence of prediabetes was 11.85% (95% CI 11.71 to 11.99). Follow-up data were available for 23,293 individuals with prediabetes. Among them, 36.08% (95% CI 35.46 to 36.70) returned to normoglycemia, 40.92% (95% CI 40.29 to 41.55) had persistent prediabetes, and 23.00% (95% CI 22.46 to 23.54) progressed to T2D. The risk for persistence of prediabetes and for progression to T2D increased with age, body mass index (BMI), triglyceride level, and less than 150 min/week of physical activity. An HbA1c level of 6% or greater was the strongest individual predictor of progression to T2D. Conclusions: Physical activity, diet, smoking, and BMI are modifiable factors that are associated with the persistence of prediabetes and the progression to T2D. The workplace is a feasible setting for the early detection of prediabetes and the promotion of lifestyles that can prevent progression to T2D.

## 1. Introduction

Diabetes is a global public health problem whose prevalence has steadily increased during recent decades, and is now one of the main causes of morbidity and mortality in adults [1,2]. The International Diabetes Federation reported that the global prevalence of diabetes is currently 9.3% (463 million people) and estimated that the prevalence in 2045 will be 10.9% (700 million people) [3]. Furthermore, approximately 50% of all people with diabetes worldwide are undiagnosed [4]. Type 2 diabetes (T2D), the most common type of diabetes, is a lifestyle-related condition that can be delayed or prevented by appropriate interventions. 

T2D is commonly an asymptomatic disease [5] that is associated with cardiovascular diseases, cancer, dental diseases, a large number and duration of hospitalizations [6], lower quality of life, and reduced life expectancy [7]. Diabetes has an enormous economic impact on public health systems and on individuals and their families [4]. 

People whose blood glucose levels are higher than normal but do not fulfill the criteria for a diagnosis of T2D are considered to have prediabetes, also known as dysglycemia or intermediate hyperglycemia [8]. Identification of these individuals is important because they have a high risk of developing T2D [9]. In particular, these individuals have similar cardiovascular risk as people with diabetes, and also have an increased risk of retinopathy, kidney disease, and neurological disorders [10]. 

Diagnostic criteria of prediabetes is based on biochemical parameters, and include at least one of the following conditions: impaired fasting plasma glucose (FPG), impaired glucose tolerance (IGT), and altered glycated hemoglobin (HbA1c) [11,12]. Epidemiological studies showed a lack of agreement between HbA1c and glucose-based tests (FPG and IGT) in defining prediabetes [13,14]. Moreover, there is currently no consensus regarding the diagnostic criteria for prediabetes. The American Diabetes Association (ADA) defined prediabetes as a FPG of 100 to 125 mg/dL [9], but the National Institute for Health and Care (NICE), the International Expert Committee (IEC), and the World Health Organization (WHO) defined prediabetes as an FPG of 110 to 125 mg/dL [12,15]. Regarding the HbA1c level, the ADA recommends a cutoff of 5.7 to 6.4%, the NICE and IEC recommend a cutoff of 6.0% to 6.4% [16], and the WHO concludes with a lack of evidence to make a formal recommendation on the interpretation of HbA1c levels below 6.5% [17]. The WHO criteria have the lowest false positive rate for progression to T2D, and the ADA criteria have the lowest false negative rate and the highest sensitivity [18]. Notably, according to the ADA criteria, one-third of the worldwide population has prediabetes, even though many of them only have a low risk of morbidity and mortality [19,20]. 

The prevalence of prediabetes (27% to 49%, depending on the diagnostic criteria) [20] differs by age, gender, ethnicity, region of residence, and socioeconomic status [3]. As noted above, there are some difficulties in determining the prevalence of T2D because of the different criteria. FPG is widely accepted as a clinical diagnostic criterion with a good cost-efficiency ratio, and HbA1c is considered an accurate alternative method that has better predictive capacity but higher cost [21].

The rate of progression from prediabetes to diabetes is about 5% to 10% annually [11,22,23] and 70% of individuals with prediabetes will develop diabetes during their lifetimes, with a greater risk for those who are overweight or obese [24]. Few studies have evaluated the influence of different risk factors on the progression to T2D. A family history of diabetes, obesity, low level of high-density lipoprotein cholesterol (HDL-C), high systolic blood pressure (SBP), general and abdominal obesity, tobacco smoking, gestational diabetes, and ethnicity are associated with progression to T2D [7,8,25]. However, the importance of diet and physical activity (PA) are uncertain, and the interactions among different risk factors is a topic of active investigation [25,26]. Follow-up studies have shown that many people with prediabetes revert to normoglycemia, and that this is accompanied by changes in certain cardiovascular risk factors [8,27,28]. 

The main aim of the present study was to describe the incidence of T2D in a large cohort of workers from a south-European Mediterranean population who have prediabetes, and to evaluate sociodemographic, clinical, metabolic, and lifestyle factors that affect the risk for the persistence of prediabetes and the progression to T2D.

## 2. Materials and Methods 

The present study examined a cohort of 23,293 working adults from Spain who had prediabetes (aged 20 to 65 years) and who worked in service industries, construction, public administration, healthcare, or postal services. Participants were selected from a total population of 234,995 potentially eligible individuals who received occupational medical examinations between 2012 and 2013. The inclusion criteria were: age between 20 and 65 years and FPG between 100 and 125 mg/dL, according to the ADA criteria [9]. Individuals were excluded if they had a history of physician-diagnosed diabetes, used an oral antidiabetic drug or systemic glucocorticoid, had been treated for cancer during the preceding 5 years, had anemia (haematocrit < 36% in men and < 33% in women), were pregnant, or had an FPG of 126 mg/dL or more or an HbA1c of 6.5% or more at baseline. All participants completed standard health examinations, anthropometric measurements, and metabolic tests at baseline. Follow-up examinations were performed in 2017 and 2018.

The study protocol was in accordance with the Declaration of Helsinki and was approved by the Ethics Committee of the Balearic Islands Health Service (CEI-IB Ref. No: 1887). Participants were informed of the purpose of the study before they provided consent to participate.

### 2.1. Data Collection and Definition of Variables

Sociodemographic characteristics, clinical data, and lifestyle data were collected at the baseline assessment. Participants were asked to report if they engaged in moderate and/or vigorous PA (at least 150 min/week, following WHO recommendations) and if they consumed fruits and vegetables daily; the responses to these two questions were recorded as “yes” or “no”. Each individual was also categorized as a smoker, former smoker, or never smoker, and as a “blue collar” or “white collar” worker using the Spanish Society of Epidemiology classification, which is based on occupational social status, a classification that has good reliability and a high correlation with educational level [29].

All anthropometric measurements were made according to the recommendations of the International Standards for Anthropometric Assessment (ISAK) and were performed by well-trained technicians [30]. Body weight was measured to the nearest 0.1 kg using an electronic scale. Height was measured to the nearest 0.5 cm using a stadiometer. Body mass index (BMI) was calculated as weight divided by height-squared (kg/m^2^). Criteria used to define overweight were the ones of the WHO, which considers overweight when BMI ≥25 to 29,9 kg/m^2^ and obesity when BMI ≥30 kg/m^2^.

Venous blood samples were taken from an antecubital vein with suitable vacutainers (without an anticoagulant to obtain serum) in the morning after a 12 h overnight fast. The concentrations of glucose, HbA1c, and triglycerides were measured by standard procedures using an automated system (Beckman Coulter SYNCHRON CX^®^ 9 PRO, Brea, CA, USA).

Blood pressure was determined after a resting period of 10 min in a supine position using an electric and calibrated sphygmomanometer; 3 measurements were performed at 1 min intervals and the average was calculated. 

Hypertension was defined as a SBP of 140 mmHg or more, or a diastolic blood pressure (DBP) of 90 mmHg or more, or taking antihypertensive medication. High triglycerides (TG) was defined as 150 mg/dL or more, and high cholesterol was defined as 200 mg/dL or more. 

Incident T2D was defined as an FPG of 126 mg/dL or more or use of an anti-hyperglycemic medication at the follow-up assessment.

### 2.2. Statistical Analysis

Continuous variables are expressed as mean (± standard deviation (SD)) and categorical variables as n (%). The analysis of variance (ANOVA) technique was used for comparisons of continuous variables and Bonferroni post hoc analysis was used for performing post hoc comparisons. The χ^2^ test for comparisons was used for categorical variables and post hoc methods based on standardized residuals with Bonferroni correction were performed for multiple comparisons. 

Multinomial logistic regression analyses were used to calculate adjusted odds ratios (aORs) for the persistence of prediabetes, the development of T2D, and regression to normoglycemia. 

Receiver operating characteristic (ROC) curves were evaluated to compare the diagnostic performance of HbA1c, FPG, and a combination of independently associated variables (HbA1c, age, sex, social status, PA, smoking, and TG) in predicting progression to T2D. 

All analyses were performed using Statistical Package for Social Science (SPSS) version 25.0 (IBM Company, New York, NY, USA) and STATA version 10 (StataCorp, TX, USA) for Windows. All statistical tests were two-sided, and a *p* value below 0.05 was considered significant.

## 3. Results

A total of 234,995 individuals who received occupational medical examinations agreed to participate. A total of 33,355 of these individuals (14.19%, 95% confidence interval (CI) 14.05 to 14.33) had T2D, 6956 of whom (3.01%, 95% CI 2.89 to 3.03) had not been previously diagnosed with T2D (Table 1). The prevalence of prediabetes was 11.85% (95% CI 11.71 to 11.99). Relative to individuals with normoglycemia, those with diabetes or prediabetes were significantly older; more likely to be a former smoker, male, and a blue-collar worker; have hypertension and hypertriglyceridemia; and be obese or overweight.

After excluding individuals with normoglycemia or diabetes, the final sample had 27,844 individuals with prediabetes (mean age: 44.81 ± 9.91 years, 72.97% male, and 76.68% blue-collar workers). Follow-up data at five years were available for 23,293 of these individuals. Based on FPG and ADA criteria, 36.08% of those with prediabetes (95% CI 35.46 to 36.70) returned to normoglycemia, 40.92% (95% CI 40.29 to 41.55) had persistent prediabetes, and 23.00% (95% CI 22.46 to 23.54) progressed to T2D (Figure 1). This corresponded to a 4.6% annual rate of progression to T2D. The individuals in these three groups (normalized, persisted, progressed) had significant differences in age, sex, social status, PA, diet, smoking status, blood pressure, FPG, HbA1c, and TG (Table 2).

An adjusted multinomial logistic regression model (with the normalized group as reference) indicated that most of the evaluated factors had significant and independent positive or negative associations with persistence in prediabetes and progression to T2D (Figure 2). For persistence of prediabetes, the statistically significant factors were age (aOR = 1.010, 95% CI 1.007 to 1.014), BMI (aOR = 1.054, 95% CI 1.041 to 1.067), TG level (aOR = 1.001, 95% CI 1.001 to 1.002), performance of PA (aOR = 0.338, 95% CI 0.306 to 0.373), daily consumption of fruits and vegetables (aOR = 0.742, 95% CI 0.681 to 0.810), status as former smoker (aOR = 1.714, 95% CI 1.550 to 1.895), and status as a current smoker (aOR = 1.124, 95% CI 1.047 to 1.208). 

For progression to T2D, the statistically significant factors were age (aOR = 1.123, 95% CI 1.114 to 1.132), BMI (aOR = 1.860, 95% CI 1.813 to 1.908), TG level (aOR = 1.002, 95% CI 1.001 to 1.002), performance of PA (aOR = 0.059, 95% CI 0.040 to 0.086), status as former smoker (aOR = 3.291, 95% CI 2.720 to 3.981), and status as a never smoker (aOR = 1.158, 95% CI 1.003 to 1.336). Three factors were independently associated with progression to T2D but not persistence of prediabetes: HbA1c below 6% (aOR = 0.016, 95% CI 0.013 to 0.019), blue-collar social status (aOR = 0.727, 95% CI 0.624 to 0.849), and male gender (aOR = 1.221, 95% CI 1.050 to 1.420). 

An alternative multinomial logistic regression model with the persistence in prediabetes as a reference group showed similar results for progression, except for current smoking, that was not statistically significant. Daily consumption of fruits and vegetables, performance of PA (≥150 min/week) and HbA1c below 6% were significantly associated with regression to normoglycemia; while, age, male sex, SBP, TG and former smoker were significantly associated with persistence in prediabetes, as shown in the supplementary material (Appendix A).

Analysis of participants who fulfilled the ADA criteria for prediabetes (FPG: 100 to 109 mg/dL and HbA1c <6.0%) indicated that the annual rate of progression to T2D was 0.31% (193 of 12,485 individuals). For those who met the more restrictive WHO criteria for prediabetes (46.40% of the sample) the annual rate of progression to T2D was 9.56% (5165 of 10,808 individuals; data not shown).

Our results indicated the HbA1c level was the strongest predictor of progression to T2D. In particular, participants who progressed to T2D had significantly higher levels of HbA1c than those who did not (6.20% ± 0.16 vs. 5.89% ± 0.18, *p* < 0.001). Our ROC analysis (Figure 3) showed that the FPG level had limited value for prediction of progression to T2D (area under the curve (AUC) = 0.632, 95% CI 0.623 to 0.640). In contrast, HbA1c level had good predictive performance (AUC = 0.882, 95% CI 0.877 to 0.887). Furthermore, a model that considered age, sex, BMI, TG, smoking status, PA, and HbA1c had very high predictive capacity (AUC = 0.977, 95% CI 0.975 to 0.979).

## 4. Discussion

This large cohort study identified risk factors for the progression from prediabetes to T2D among workers over a period of 5 years. Nearly 1 of every 4 people with prediabetes progressed to T2D, and 1 of 3 reverted to normoglycemia. Our results showed that physical inactivity, high BMI, and hypertriglyceridemia independently increased the risk for T2D and persistence of prediabetes. HbA1c was the strongest single predictor of T2D progression among those with impaired FPG. 

The baseline prevalence of prediabetes in our population of workers was lower than reported in a meta-analysis of middle-age participants that used the WHO and ADA criteria for diagnosis of prediabetes [20]. Prevalence estimates of prediabetes vary widely within the literature, depending on the diagnostic criteria, biochemical parameters, and certain characteristics of the study population. Furthermore, a diagnosis of prediabetes based on more than one criterion may overestimate its prevalence, and this could explain some of the differences in previous estimations [28].

Our results showed a similar progression rate (4.6% annually) to T2D among Spanish workers as previously reported in multiple studies of different general populations [18,26,31,32,33,34]. The Atherosclerosis Risk in Communities (ARIC) and the Brazilian Longitudinal Study of Adult Health (ELSA-Brazil) studies reported lower progression rates (2.3% and 3.5%, respectively), but their study populations had certain unique sociodemographic characteristics. In particular, the ARIC included more women and the ELSA-Brazil included a high percentage of people who had high educational levels [18,33]. The rate of progression from prediabetes to T2D also varies according to the diagnostic criteria for prediabetes. Although some studies reported better prediction of progression to T2D using FPG rather than HbA1c for diagnosis of prediabetes [18], a recent meta-analysis that compared different definitions of prediabetes in prediction of subsequent T2D reported that HbA1c was a better predictor than FPG [35]. According to a previous study [28], we diagnosed prediabetes based on one parameter (FPG), and measured HbA1c when the FPG was elevated. Combining different definitions of prediabetes can greatly inflate the prevalence, and the use of a combination of measurements is not relevant in clinical practice [28]. Although, similar to previous studies [20,36,37], we found a very high rate of progression to T2D when both parameters were altered. 

The diagnostic criteria of prediabetes have changed over time and there is currently no worldwide consensus on the definition of this condition. Our results indicated that participants who fulfilled only the ADA criteria, but not more restrictive criteria of the WHO and IEC, had a low rate of annual progression to T2D, but those who met the IEC criteria had an annual rate of progression of about 10%. Prediabetes is a controversial term because many people with moderately high glycemic levels will not progress to T2D, and some will even experience a reversion to normoglycemia. For these reasons, the WHO discourages the use of the term “prediabetes” to prevent the stigmatization of these individuals [19].

Although persistence of prediabetes is considered preferable to progression to T2D, individuals with persistent prediabetes also have an increased risk of cardiovascular and other diseases, similar to those with T2D [6]. The present study indicated similar glucose levels among people with prediabetes and T2D at the initial assessment. However, there are probably differences in the care received by these two groups. Most patients with T2D engage in self-management and closely follow advice from a clinician or health care professional [28,38]. In contrast, most patients with prediabetes are probably not under strict glycemic control and receive less intensive care in health care settings. 

In accordance with our results, previous research also reported reversion from prediabetes to normoglycemia ranged from 20% to 50%, depending on age, gender, ethnicity, geographical region, socioeconomic status, and the criteria used to define prediabetes [8]. Moreover, a population-based observational study of the natural history of diabetes in England showed that the rate of reversion from impaired FPG to normoglycemia ranged 55% to 80% after 10 years. Several trials also reported reductions in the risk of developing T2D among individuals with prediabetes after lifestyle or therapeutic drug interventions [20,27,39,40]. Nevertheless, reversion to normoglycemia may even occur without interventions. For example, the Diabetes Prevention Program (DPP) study reported a reversion rate of 19% in the control group [23]. Similar observational studies with shorter durations (3 to 5 years follow-up) reported that the rate of return to normoglycemia was 25% [36]. The cause of reversion from prediabetes to normoglycemia requires further study, and it is uncertain whether changes in cardiovascular risk factors also correspond to a lower risk of cardiovascular disease and death [6]. 

We identified several common risk factors for the persistence of prediabetes and the progression to T2D—age, BMI, PA, smoking, HbA1c, and TG—similar to other studies [31,33,34]. A recent systematic review and meta-analysis reported that PA prevented the progression from prediabetes to T2D [41]. Another study reported that following the WHO recommendation of 150 min/week of PA could reduce FPG by 5%, reduce HbA1c by 4%, and also reduce the rate of progression to T2D [42]. Physical inactivity and obesity are the main modifiable risk factors that should be targeted for prevention of T2D and its complications. Thus, previous interventional studies that focused on lifestyle changes have shown some success in diabetes prevention [39,43]. The DPP in the U.S. and the Finnish Diabetes Prevention Study (DPS) reported a risk reduction of approximately 50% for the progression to T2D for people with prediabetes who received interventions focused on a healthy diet and PA [40,44]. Moreover, the DPP reported that lifestyle interventions reduced risk more than metformin treatment (58% vs. 38%). Lifestyle interventions that normalize glucose tolerance in people with FPG can also reduce the probability of cardiovascular diseases and mortality. 

Our study highlighted the importance of control of BMI and regular PA in preventing the progression to T2D. Thus, collection of data about lifestyle-related factors, particularly PA, in people with prediabetes is needed to estimate the risk of progression to T2D and the risk of cardiovascular events [45,46]. Self-reported and/or objective measures at the individual level should be routinely used in primary-care settings, combined with anthropometric measures, to advise patients and promote healthy lifestyles changes [47]. 

Although we found that white-collar workers had a higher rate of progression to T2D, the baseline prevalence of T2D was lower for this group of workers. One possible explanation for this is that we only included people with prediabetes, and prediabetes is less common among white-collar workers. Thus, this is an example of conditioning on a variable (prediabetes) that was affected by exposure (social status). As previously described [48], this could lead to selection bias and a spurious association between exposure and outcome (in our case, social status and progression to T2D). Another study also described this paradox in an evaluation of the association between BMI and mortality in women with diabetes [49].

Some limitations of our study should be considered. First, we included data from periodic health examinations at the workplace, and HbA1c was only collected from subjects with previously identified elevation of FPG. Furthermore, we did not perform an oral glucose tolerance test (OGTT). In general, the OGTT is more sensitive but less specific than fasting glucose for identification of people who will develop diabetes [50]. However, the low reproducibility, high cost, and prolonged time required for this test have limited its use in clinical practice [51]. Also, there is a great possibility of regression toward the mean phenomenon taking place and possibly affecting the regression rate. Finally, we did not assess changes in lifestyle (diet, PA, etc.) of study subjects during the 5-year follow-up, and this could have led to misclassification bias.

The main strengths of our study are the large sample size and the relatively long follow-up time. We examined participants from multiple occupations and regions of the country, and our study population may be considered representative of Spanish workers. Further studies evaluating the reversion from prediabetes to normoglycemia and the role of lifestyle and clinical outcomes are required to develop a universally accepted definition of prediabetes and to identify individuals most suitable for interventions.

## 5. Conclusions

This study assessed the progression from prediabetes to T2D in a large and representative sample of workers from Spain. Although our results indicated that some of these individuals progressed to T2D, most of them remained prediabetic or reverted to normoglycemia. Our study suggests that control of the BMI and regular PA may help prevent the progression to T2D. 

It is crucial to identify individuals with prediabetes who would benefit most from preventive strategies so that they better understand the risk of progression to T2D and can take appropriate preventative measures. The workplace is a feasible setting for the early detection of prediabetes, assessment of the risk of progression to T2D, and identification of subjects who can benefit most from lifestyle changes.

## Figures and Tables

**Figure 1 nutrients-12-01538-f001:**
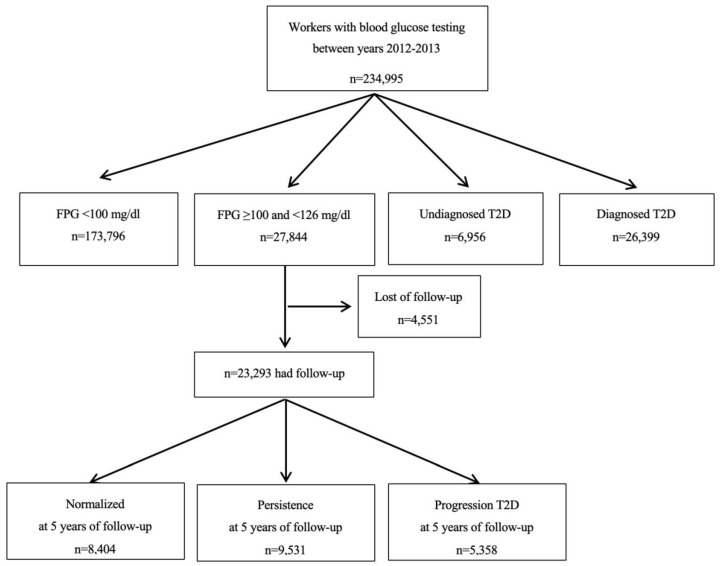
Flow chart for study participants.

**Figure 2 nutrients-12-01538-f002:**
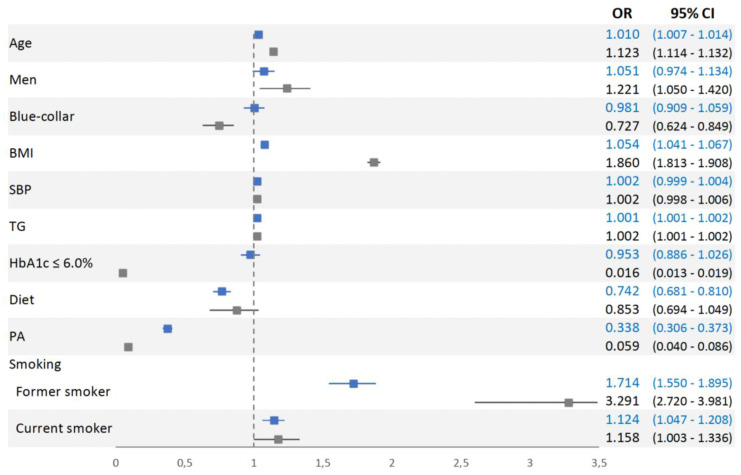
Multinomial logistic regression analysis of participants who persisted in prediabetes (blue) and progressed to T2D (grey) relative to those who returned to normoglycemia (reference group) after 5 years of follow-up. BMI, body mass index; SBP, systolic blood pressure; TG, triglycerides; HbA1c, glycated hemoglobin; Diet (daily consumption of fruits and vegetables); PA, physical activity (≥150 min/week).

**Figure 3 nutrients-12-01538-f003:**
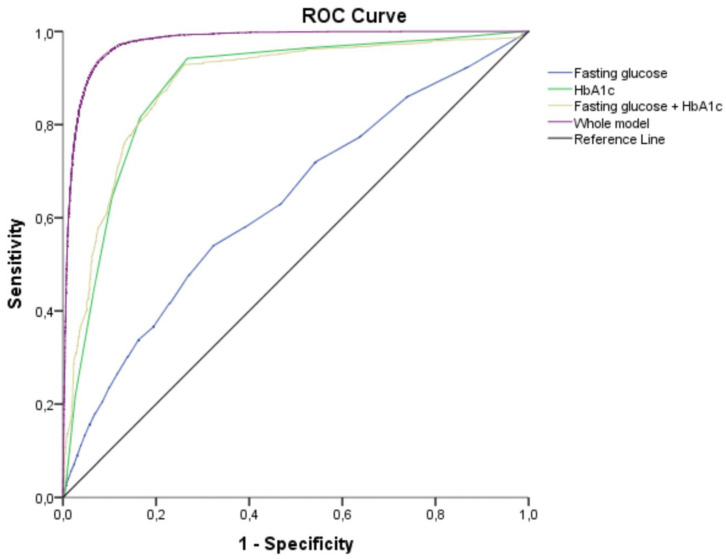
Receiver operating characteristic curves for the prognostic value of different variables in predicting progression from prediabetes to diabetes after 5 years. FPG: area under the curve (AUC) = 0.632 (95% CI 0.623 to 0.640); HbA1c: AUC = 0.882 (95% CI 0.877 to 0.887); FPG + HbA1c: AUC = 0.884 (95% CI 0.879 to 0.889), age, sex, BMI, TG, smoking status, PA, and HbA1c: AUC = 0.977 (95% CI 0.975 to 0.979).

**Table 1 nutrients-12-01538-t001:** Characteristics of the total population at assessment for eligibility (*n* = 234,995).

Variable	Normoglycemia FPG < 100 mg/dL (*n* = 173,796)	Prediabetes FPG 100–125 mg/dL (*n* = 27,844)	Diabetes FPG ≥ 126 mg/dL (*n* = 33,355)	*p* Value
Sex (male)	95,414 (54.9%)	20,131 (72.3%)	21,781 (65.3%)	<0.001
Age (years)	38.07 ± 10.19	44.62 ± 10.04	47.38 ± 9.23	<0.001
Social status Blue-collar	122,700 (70.6%)	21,329 (76.6%)	25,016 (75.0%)	<0.001
BMI (kg/m^2^)	25.20 ± 4.26	27.79 ± 4.82	29.69 ± 5.02	<0.001
BMI categories				
Overweight	57,874 (33.3%)	11,973 (43.0%)	15,710 (47.1%)	<0.001
Obesity	21,725 (12.5%)	7713 (27.7%)	13,642 (40.9%)	
WC (cm)	82.10 ± 10.47	87.01 ± 10.48	87.67 ± 10.58	<0.001
SBP (mmHg)	120.19 ± 15.32	127.65 ± 17.09	130.92 ± 17.94	<0.001
DBP (mmHg)	73.04 ± 10.45	78.17 ± 11.18	80.14 ± 11.46	<0.001
Hypertension	25,374 (14.6%)	8214 (29.5%)	12,541 (37.6%)	<0.001
FPG (mg/dL)	84.62 ± 9.39	106.24 ± 5.86	109.25 ± 41.63	<0.001
TG (mg/dL)	102.40 ± 62.92	136.64 ± 104.10	147.81 ± 101.04	<0.001
High TG	9733 (5.6%)	3954 (14.2%)	5537 (16.6%)	<0.001
Total Cholesterol (mg/dL)	188.51 ± 36.19	203.63 ± 38.60	202.78 ± 36.26	<0.001
High Cholesterol	14,773 (8.5%)	3487 (16.5%)	4670 (14.0%)	<0.001
Smoking status				
Former smoker	15,642 (9.0%)	4260 (15.3%)	7772 (23.3%)	<0.001
Current smoker	59,438 (34.2%)	9189 (33.0%)	9306 (27.9%)	

Results are given in mean ± standard deviation (SD) or n (%); BMI, body mass index; WC, waist circumference; SBP, systolic blood pressure; DBP, diastolic blood pressure; FPG, fasting plasma glucose; TG, triglycerides.

**Table 2 nutrients-12-01538-t002:** Baseline characteristics of participants with prediabetes who had 5-year follow-up data (*n* = 23,293) and experienced normoglycemia, persistent prediabetes, or progression to T2D.

Variables	Normalized (*n* = 8404)	Persisted (*n* = 9531)	Progressed (*n* = 5358)	*p* Value
Age (years)	42.14 ± 10.14	44.41 ± 9.47	48.65 ± 8.78	<0.001^*,‡,†^
Male (%)	5925 (70.5%)	7154 (75.1%)	3913 (73.0%)	<0.001^*^
Blue-collar	7942 (94.5%)	9050 (95.0%)	5146 (96.0%)	0.130
Smoking status				
Former smoker	910 (10.8%)	1606 (16.9%)	1041 (19.4%)	<0.001^*,‡,†^
Current smoker	3034 (36.1%)	3294 (34.6%)	1407 (26.3%)	
PA (≥150 min/week)	5777 (68.7%)	2958 (31.0%)	61 (1.1%)	<0.001^*,‡,†^
Diet (daily fruits and vegetables)	5170 (61.5%)	2968 (31.1%)	403 (7.5%)	<0.001^*,‡,†^
BMI (kg/m^2^)	25.09 ± 3.24	26.98 ± 3.58	33.42 ± 4.09	<0.001^*,‡,†^
FPG (mg/dL)	104.65 ± 4.73	106.37 ± 5.79	108.50 ± 6.73	<0.001^*,‡,†^
HbA1c (%)	5.892 ± 0.19	5.89 ± 0.17	6.20 ± 0.16	<0.001^*,‡,†^
SBP (mmHg)	123.76 ± 16.21	127.25 ± 16.25	134.72 ± 17.5	<0.001^*,‡,†^
DBP (mmHg)	75.39 ± 10.69	78.13 ± 10.78	82.75 ± 11.1	<0.001^*,‡,†^
TG (mg/dL)	108.56 ± 72.55	145.32 ± 116.72	166.84 ± 112.79	<0.001^*,‡,†^

Results are given as mean ± SD or n (%); PA, physical activity; HbA1c, glycated hemoglobin. * Statistically significant differences between normalized vs. persisted (‡) normalized vs. progressed and (†) persisted vs. progressed (Bonferroni post hoc analysis).

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
