# Peer review of "Lifestyle and Progression to Type 2 Diabetes in a Cohort of Workers with Prediabetes"

_nutrients, 2020, doi:10.3390/nu12051538_

Round 1

Reviewer 1 Report

The manuscript by Benassar-Veny et al aimed to analyze a number of factors which may contribute to progression to type 2 diabetes (T2D) or to reversion to normoglycemia in a large cohort of prediabetic subjects. 234995 individuals, who received occupational medical examinations, were classified as normoglycemic (Fasting Plasma glucose-FPG<100 mg/dl), pre-diabetic (FPG 100-125 mg/dl) and diabetic (FPG≥ 126 mg/dl). Relative to normoglycemic group, the other 2 groups had significantly higher age, BMI, WC, SBP, DBP, TG and cholesterol. Moreover, there were more smokers, males and blu-collar workers. Pre-diabetic subjects were 27844 and, of those, 23293 had a 5-year follow-up. Among this group, 8404 subjects reversed to normoglycemia (normalized group), 9531 remained prediabetic (persisted group) and 5358 progressed to T2D (progressed group). The overall p value in these groups was different for all the variables examined (age, sex, occupational status, smoke, physical activity-PA, diet, BMI, HbA1c, SBP, DBP, TG). A logistic regression model, with the normalized group as reference, indicated that  most variables had significant and independent associations with the pre-diabetes or T2D. Finally, the authors showed that a model that considered age, sex, BMI, TG, smoke, PA and HbA1c had a very high predictive capacity (AUC=0.977).

The manuscript is interesting and deals with an issue yet not fully clear, the progression from prediabetes to diabetes. Moreover, it is well written.

Comments:

  1. In table 2 the authors could add a multiple comparison test among the groups
  2. The logistic regression model has been performed with the normalized group as reference. It could be very interesting to analyze the variables with the persisted group as reference, in order to assess the factors associated with the T2D and with the reversion to normoglycemia (in line with the aim of the study).

Author Response

Response to Reviewer 1 Comments

The manuscript by Benassar-Veny et al aimed to analyze a number of factors which may contribute to progression to type 2 diabetes (T2D) or to reversion to normoglycemia in a large cohort of prediabetic subjects. 234995 individuals, who received occupational medical examinations, were classified as normoglycemic (Fasting Plasma glucose-FPG<100 mg/dl), pre-diabetic (FPG 100-125 mg/dl) and diabetic (FPG≥ 126 mg/dl). Relative to normoglycemic group, the other 2 groups had significantly higher age, BMI, WC, SBP, DBP, TG and cholesterol. Moreover, there were more smokers, males and blu-collar workers. Pre-diabetic subjects were 27844 and, of those, 23293 had a 5-year follow-up. Among this group, 8404 subjects reversed to normoglycemia (normalized group), 9531 remained prediabetic (persisted group) and 5358 progressed to T2D (progressed group). The overall p value in these groups was different for all the variables examined (age, sex, occupational status, smoke, physical activity-PA, diet, BMI, HbA1c, SBP, DBP, TG). A logistic regression model, with the normalized group as reference, indicated that most variables had significant and independent associations with the pre-diabetes or T2D. Finally, the authors showed that a model that considered age, sex, BMI, TG, smoke, PA and HbA1c had a very high predictive capacity (AUC=0.977).

The manuscript is interesting and deals with an issue yet not fully clear, the progression from prediabetes to diabetes. Moreover, it is well written.

Point 1: In table 2 the authors could add a multiple comparison test among the groups.

Response 1: Thank you ever so much for your comments. We think that they can help to improve the paper. As you suggest we have added a multiple comparison test among the groups in Table 2 (post hoc analysis). Also, we have added this information in the statistical analysis in the methods section (page 3-4, lines 147-152).

Point 2: The logistic regression model has been performed with the normalized group as reference. It could be very interesting to analyze the variables with the persisted group as reference, in order to assess the factors associated with the T2D and with the reversion to normoglycemia (in line with the aim of the study).

Response 2: Many thanks for the suggestion. We have provided a supplementary material (Table S1) with multinomial logistic regression model with the persistence in prediabetes as a reference group. We have also added a paragraph with these results in the results section (page 7, paragraph 1).

Reviewer 2 Report

The manuscript is to be thoroughly checked for the grammatical errors, sentence formation as well as spellings.

Detail Report:

 Lifestyle and progression to type 2 diabetes in a cohort of workers with prediabetes

Abstract: It is to the point, well written and connected. The aim of the study was to determine the incidence of T2D in a large cohort of workers with prediabetes, and to evaluate the influence of sociodemographic, clinical, metabolic, and lifestyle factors that affect the persistence of prediabetes and the progression to T2D.

Introduction: Explains all the aspects of prediabetes and its progression to type two diabetes. It is explained in synchronization; definition, symptoms, prone conditions, biochemical parameters including glucose levels and diagnostics.

Material and Methods:

The present study examined a cohort of 23,293 working adults from Spain who had prediabetes (aged 20 to 65 years) and who worked in service industries, construction, public administration, healthcare, or postal services. The inclusion criteria were: age between 20 and 65 years and FPG between 100 and 125 mg/dL, according to the ADA criteria

Data collection and definition of variables

Sociodemographic characteristics, clinical data, and lifestyle data were collected at the baseline assessment.

All anthropometric measurements were made according to the recommendations of the International Standards for Anthropometric Assessment (ISAK).

Body mass index (BMI) was calculated as weight divided by height-squared (kg/m2). Criteria used to define overweight were the ones of the WHO, which considers overweight when BMI ≥25 to 29,9 kg/m2 and obesity when BMI ≥30 kg/m2

Hypertension was defined as a systolic blood pressure (SBP) of 140 mmHg or more, or a diastolic blood pressure (DBP) of 90 mmHg or more, or taking antihypertensive medication. High TG was defined as 150 mg/dL or more, and high cholesterol was defined as 200 mg/dL or more.

All analyses were performed using Statistical Package for Social Science (SPSS) version 25.0

Results:

A total of 33,355 of these individuals (14.19%, 95% CI 14.05 to 14.33) had T2D, 6,956 of 158 whom (3.01%, 95% CI 2.89 to 3.03) were not previously diagnosed with T2D. The prevalence of prediabetes was 11.85% (95% CI 11.71 to 11.99). Relative to individuals with normoglycemia, those with diabetes or prediabetes were significantly older; more likely to be a former smoker, male, and a blue-collar worker; have hypertension and hypertriglyceridemia; and be obese or overweight.

Selected variables; Sex (male), Age (years), Social status, Blue-collar, BMI (Kg/m2), BMI categories, Overweight, Obesity, WC (cm), SBP (mmHg), DBP (mmHg), Hypertension, FPG (mg/Dl), TG (mg/Dl), High TG, Total Cholesterol (mg/Dl).

After excluding individuals with normoglycemia or diabetes, the final sample had 27,844 individuals with prediabetes (mean age: 44.81 ± 9.91 years, 72.97% male, and 76.68% blue-collar workers). Follow-up data at five years were available for 23,293 of these individuals. Based on FPG  and ADA criteria, 36.08% of those with prediabetes (95% CI 35.46 to 36.70) returned to  normoglycemia, 40.92% (95% CI 40.29 to 41.55) had persistent prediabetes, and 23.00% (95% CI 22.46 170 to 23.54) progressed to T2D.
Selected variables: Age (years), Male (%), Blue collar, Smoking status, Former smoker, Current smoker, Physical activity (>150 min/week), Diet (daily fruits and vegetables), BMI (kg/m2), FPG (mg/dL), HbA1c (%), SBP (mmHg) and DBP (mmHg).

Results indicated the HbA1c level was the strongest predictor of progression to T2D. In particular, participants who progressed to T2D had significantly higher levels of HbA1c than those who did not (6.20% ± 0.16 vs. 5.89% ± 0.18, p < 0.001). Repetitive Operating Characteristics analysis showed that FPG level had limited value for prediction of progression to T2D (AUC = 0.632, 95% CI 0.623 to 0.640). In contrast, HbA1c level had good predictive performance (AUC = 0.882, 95% CI 0.877 to 0.887). Furthermore, a model that considered age, sex, BMI, TG, smoking status, PA, and HbA1c had very high predictive capacity (AUC = 0.977, 95% CI 0.975 to 0.979).

Conclusion

 This study assessed the progression from prediabetes to T2D in a large and representative sample of workers from Spain. Although the results indicated that some of these individuals progressed to T2D, most of them remained prediabetic or reverted to normoglycemia. The study suggests that control of the BMI and regular Physical activity may help prevent the progression to T2D.

Author Response

Response to Reviewer 2 Comments

Point 1: The manuscript is to be thoroughly checked for the grammatical errors, sentence formation as well as spellings.

Response 1: Many thanks for the suggestion. We agree with the reviewer, we have carefully revised our manuscript. Furthermore, the whole manuscript has been cheeked for the grammatical errors, sentence formation as well as spellings and corrected for English grammar mistakes.

Point 2: Detail Report:

Abstract: It is to the point, well written and connected. The aim of the study was to determine the incidence of T2D in a large cohort of workers with prediabetes, and to evaluate the influence of sociodemographic, clinical, metabolic, and lifestyle factors that affect the persistence of prediabetes and the progression to T2D.

Introduction: Explains all the aspects of prediabetes and its progression to type two diabetes. It is explained in synchronization; definition, symptoms, prone conditions, biochemical parameters including glucose levels and diagnostics.

Material and Methods: 

The present study examined a cohort of 23,293 working adults from Spain who had prediabetes (aged 20 to 65 years) and who worked in service industries, construction, public administration, healthcare, or postal services. The inclusion criteria were: age between 20 and 65 years and FPG between 100 and 125 mg/dL, according to the ADA criteria

Data collection and definition of variables

Sociodemographic characteristics, clinical data, and lifestyle data were collected at the baseline assessment.

All anthropometric measurements were made according to the recommendations of the International Standards for Anthropometric Assessment (ISAK).

Body mass index (BMI) was calculated as weight divided by height-squared (kg/m2). Criteria used to define overweight were the ones of the WHO, which considers overweight when BMI ≥25 to 29,9 kg/m2 and obesity when BMI ≥30 kg/m2

Hypertension was defined as a systolic blood pressure (SBP) of 140 mmHg or more, or a diastolic blood pressure (DBP) of 90 mmHg or more, or taking antihypertensive medication. High TG was defined as 150 mg/dL or more, and high cholesterol was defined as 200 mg/dL or more.

All analyses were performed using Statistical Package for Social Science (SPSS) version 25.0

Results:

A total of 33,355 of these individuals (14.19%, 95% CI 14.05 to 14.33) had T2D, 6,956 of 158 whom (3.01%, 95% CI 2.89 to 3.03) were not previously diagnosed with T2D. The prevalence of prediabetes was 11.85% (95% CI 11.71 to 11.99). Relative to individuals with normoglycemia, those with diabetes or prediabetes were significantly older; more likely to be a former smoker, male, and a blue-collar worker; have hypertension and hypertriglyceridemia; and be obese or overweight.

Selected variables; Sex (male), Age (years), Social status, Blue-collar, BMI (Kg/m2), BMI categories, Overweight, Obesity, WC (cm), SBP (mmHg), DBP (mmHg), Hypertension, FPG (mg/Dl), TG (mg/Dl), High TG, Total Cholesterol (mg/Dl).

After excluding individuals with normoglycemia or diabetes, the final sample had 27,844 individuals with prediabetes (mean age: 44.81 ± 9.91 years, 72.97% male, and 76.68% blue-collar workers). Follow-up data at five years were available for 23,293 of these individuals. Based on FPG  and ADA criteria, 36.08% of those with prediabetes (95% CI 35.46 to 36.70) returned to  normoglycemia, 40.92% (95% CI 40.29 to 41.55) had persistent prediabetes, and 23.00% (95% CI 22.46 170 to 23.54) progressed to T2D.
Selected variables: Age (years), Male (%), Blue collar, Smoking status, Former smoker, Current smoker, Physical activity (>150 min/week), Diet (daily fruits and vegetables), BMI (kg/m2), FPG (mg/dL), HbA1c (%), SBP (mmHg) and DBP (mmHg).

Results indicated the HbA1c level was the strongest predictor of progression to T2D. In particular, participants who progressed to T2D had significantly higher levels of HbA1c than those who did not (6.20% ± 0.16 vs. 5.89% ± 0.18, p < 0.001). Repetitive Operating Characteristics analysis showed that FPG level had limited value for prediction of progression to T2D (AUC = 0.632, 95% CI 0.623 to 0.640). In contrast, HbA1c level had good predictive performance (AUC = 0.882, 95% CI 0.877 to 0.887). Furthermore, a model that considered age, sex, BMI, TG, smoking status, PA, and HbA1c had very high predictive capacity (AUC = 0.977, 95% CI 0.975 to 0.979).

Conclusion

This study assessed the progression from prediabetes to T2D in a large and representative sample of workers from Spain. Although the results indicated that some of these individuals progressed to T2D, most of them remained prediabetic or reverted to normoglycemia. The study suggests that control of the BMI and regular Physical activity may help prevent the progression to T2D.

Response 2: Thank you ever so much for your detailed review.

Round 2

Reviewer 1 Report

The authors have satisfactorily addressed my previous concerns